# Liver transplant recipients with polycystic liver disease have longer waiting times but better long-term clinical outcomes than those with liver disease due to other causes: A retrospective cross-sectional study

**Matt Gittus**[1]*, **Joanna Moore**[2‡], **Albert C. M. Ong**[1‡]

**1** Academic Nephrology Unit, Division of Clinical Medicine, Faculty of Health, Sheffield Teaching Hospitals Trust, University of Sheffield and Sheffield Kidney Institute, Sheffield, United Kingdom, **2** Liver Transplant Unit, Leeds Teaching Hospitals Trust, Leeds, United Kingdom

‡ JM and ACMO are joint senior authors on this work.
* m.gittus@sheffield.ac.uk

**Data Availability Statement:** The data underlying the results presented in the study are available

## Abstract

### Introduction

Liver transplantation is the only curative option for patients with polycystic liver disease (PLD). In the United Kingdom, these patients are listed on the variant syndrome list due to their preserved liver function reflected in the United Kingdom End-stage Liver Disease (UKELD) score. The transplantation and survival rates for this patient group in the UK have not been previously reported.

### Methods

A retrospective cross-sectional analysis of patients receiving liver transplantation between 2010 and 2017 was performed using the NHS blood and transplantation database. This database contains the demographic, clinical parameters, indication for transplantation and follow-up of all patients in UK-based transplant centres. Basic statistics was performed using SPSS version 27.

### Results

5412 recipients received elective liver allografts in the study period. 1.6% (100) of recipients had PLD as their primary indication for transplantation with 60 receiving liver only allografts and 40 receiving combined liver-kidney allografts. PLD patients had a >3-fold longer mean waiting time for transplantation compared to non-PLD patients, 508 days v 154 days respectively. PLD patients receiving combined liver-kidney allografts had a longer waiting time than those receiving a liver only allograft, 610 days v 438 days respectively. There were comparable patient survival rates for people with PLD and non-PLD primary indications at 30 days (94.0% vs 97.6%) and 1 year (92.0% vs 93.2%) but improved survival rates at 5 years (81.3% vs 76.5%). There were also comparable allograft survival rates for people with

from NHS Blood and Transplant (statistical.
enquiries@nhsbt.nhs.uk).

**Funding:** The author(s) received no specific
funding for this work.

**Competing interests:** The authors have declared
that no competing interests exist.

**Abbreviations:** ADPKD, Polycystic Kidney Disease;
CLD, Chronic Liver Disease; HCC, Hepatocellular
Carcinoma; MELD, Model for End-stage Liver
Disease; NHSBT, National Health Service Blood and
Transplant; ODT, Organ Donation and
Transplantation; PLD, Polycystic Liver Disease; UK,
United Kingdom; UKELD, United Kingdom Model
for End-Stage Liver Disease; VS, Variant
Syndrome.

PLD and non-PLD primary indications at 30 days (93.9% vs 95.3%) and 1 year (91.9% vs 91.2%) but improved survival rates at 5 years (82.5% vs 77.3%). Transplant centre-level analysis identified variation in the proportion of liver transplantations for people with PLD as their primary listed indication.

## Conclusions

Patients with PLD wait significantly longer for liver transplantation compared to other indications. However, transplanted PLD patients demonstrate better longer-term patient and liver allograft survival rates compared to transplanted non-PLD patients. The unexpected variation between individual UK centres transplanting for PLD deserves further study.

## Introduction

Polycystic Liver Disease (PLD) is the most common extrarenal feature of Autosomal Dominant Polycystic Kidney Disease (ADPKD) but may occur in isolation due to other rare gene variants [1–3]. The prevalence of PLD is estimated to be 1:100,000–1:1,000,000, It is usually asymptomatic but often identified when ADPKD is diagnosed. In contrast, patients with isolated Autosomal Dominant Polycystic Liver Disease (ADPLD) often remain undiagnosed [4–7] or under-diagnosed [8]. Advanced disease may lead to abdominal pain, cyst infection, dyspnoea, anorexia and reduced quality of life [9]. The majority of these symptoms correlate with the increase in total liver volume leading to compression of adjacent tissues despite the preservation of liver function [5]. There is a striking gender imbalance with a female-to-male ratio of 6:1 despite the autosomal dominant inheritance pattern of both ADPKD and ADPLD. Females also tend to develop a more severe phenotype with higher average liver volumes and younger age of presentation [7, 10–14]. This is likely to relate to the known effects of oestrogen on liver cyst growth [15].

Liver transplantation remains the only curative option for patients with PLD, being reserved for highly symptomatic patients to relieve symptom burden and improve quality of life [16–19]. Liver function is preserved in PLD and there is no increased risk of hepatocellular carcinoma [20]. Thus, most patients would not fulfil the common criteria for transplantation listing secondary to chronic liver disease (CLD), which requires a United Kingdom model for End-stage Liver Disease (UKELD) score ≥49, or the presence of hepatocellular carcinoma. Patients with chronic liver disease whose UKELD score is <49 are referred to as having a "variant syndrome" (VS). The PLD 'variant syndrome' indications for transplantation according to NHS Blood and Transplant (NHSBT) liver selection criteria and recipient registration include "intractable symptoms due to mass of liver or pain unresponsive to cystectomy, or severe complications secondary to portal hypertension" [21]. Swenson et al reported significant symptomatic relief and improvement in quality of life after transplant at their US-based transplant centre [22].

Prior to 2018, liver allografts were allocated on a regional basis in the UK with the "local" transplant centre receiving the first offer followed by allocation to other centres by blood group compatibility, size match and greatest need [23]. In 2018, the UK changed from a regional to a national allocation scheme: specifically, this involved the DBD (donation after brain death) liver allograft being offered to a named patient. The policy change was implemented to reduce waiting list mortality. Part of this change included a proportional offering of a 90% probability of selecting the Chronic Liver Disease/Hepatocellular carcinoma (CLD/HCC) list and a 10% probability of selecting the variant syndrome list based on their relative prevalence on the waiting list. This ensured that patients with variant syndromes would have

an earlier opportunity to receive a liver transplant. In addition, patients were also ranked according to their length of time on the waiting list [21, 24].

The primary objective of this study was to compare the waiting times and outcomes of patients with PLD and those with other indications for liver transplantation prior to the change in the allocation model for the UK. A secondary objective was to identify any variation in liver transplantation rates for patients with PLD between UK based transplant centres.

## Materials and methods

A retrospective cross-sectional study was performed from the NHSBT Liver Transplant dataset of all patients undergoing primary liver transplantation in the UK between 01/01/2010 to 31/12/2017. This data is collected by the UK Transplant Registry through the DonorPath application software and electronic/paper forms. Data is submitted by NHSBT and hospital staff provide transplant follow-up information. All patient identifiers were irreversibly removed by NHSBT prior to access to the dataset and data subjects were not identifiable. Variables included the year of transplantation, CLD/HCC or VS category, cause of liver disease, waiting time, transplant centre, recipient details (gender, age and ethnicity), creatinine at the time of listing, UKELD, Model for End-stage Liver Disease (MELD) score and survival. Survival rates were informed by patient death (patient survival), allograft failure (allograft survival) or either combined (transplant failure). Before and during data analysis, the researchers were blinded to the participant's diagnosis and the transplanting centre in order to reduce bias. Data extraction occurred on 16/09/2021. A check-list has been undertaken and included according to the recommendations of STrengthening the Reporting of OBservational studies in Epidemiology (STROBE) in the S1 Checklist.

### Eligibility criteria

- Age $\geq$ 18 years

- Recipient of a liver allograft, liver only or combined liver-kidney, between 01/01/2010 and 31/12/2017

- Transplantation at a UK-based transplant centre

- Not transplanted for a super-urgent indication

The primary liver disease diagnosis is reported by the referring transplant centre. Diagnosis of PLD may be based on genetic analysis or the identification of multiple liver cysts on liver imaging (Reynolds criteria) [4].

Statistical analysis was performed using SPSS 2020 and Kaplan-Meier survival analysis. Quantitative variables were analysed in their continuous form with no categorisation performed. Any data subjects with incomplete demographic, primary disease, list type or survival data were excluded. We compared recipient characteristics using the Chi$^2$ test for categorical variables and the independent t-test for continuous variables. Sub-group analysis was performed using ANOVA to assess the difference between liver only and liver-kidney allografts. Ethical approval was not sought as the data provided was fully anonymised and routinely collected by NHSBT: Organ Donation and Transplantation (ODT).

## Results

### Study participants

All study participants included in the data analysis are summarised in Fig 1. 5412 recipients received elective liver allografts between 2010 and 2017 in UK transplant centres. Their

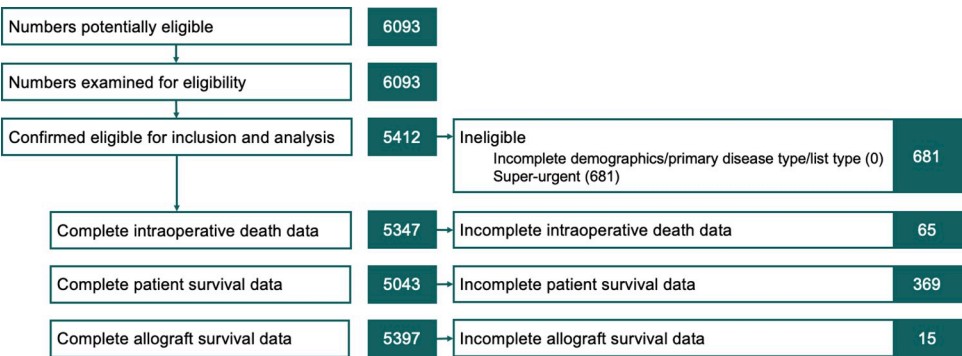

**Fig 1. Flowchart of study participants included in data analysis.**

characteristics are summarised in Table 1. 5321 (98.3%) received a liver allograft alone and the remaining 91 (1.7%) received a combined liver-kidney allograft.

100 patients (1.6%) underwent transplantation with PLD as the primary indication from the elective transplantation list. There was a clear predominance of female patients in the PLD group (78.0%) and male patients in the non-PLD group (65.7%). Despite a similar mean age at transplantation (52.1 v 52.7 years), the non-PLD group was more evenly distributed across the age groups compared to the majority of recipients being in the 48–57 age group for the PLD group (Fig 2). There was a predominance of Caucasian recipients in all categories (S1 Table). Patients with PLD had a higher baseline listing creatinine compared to recipients transplanted for non-PLD indications.

## Waiting time

Patients with PLD not meeting the criteria for the CLD/HCC indication, those listed variant syndrome pathway, had the longest waiting time for liver transplantation when compared to

**Table 1. Liver allograft recipient characteristics.**

| List type | Non-PLD | | | PLD | | | Chi-square |
|---|---|---|---|---|---|---|---|
| | CLD/HCC | VS | Total | CLD/HCC | VS | Total | |
| Number of transplants | 4988 | 324 | 5312 | 26 | 74 | 100 | - |
| Liver only | 4948 (99.2%) | 313 (96.6%) | 5261 (99.0%) | 14 (53.8%) | 46 (62.2%) | 60 (60.0%) | <0.001 |
| Liver-Kidney | 40 (0.8%) | 11 (3.4%) | 54 (1.0%) | 12 (46.2%) | 28 (37.8%) | 40 (40.0%) | |
| Gender (%) | M:3313 (66.4%) | M:177 (54.6%) F:147 (45.4%) | M:3490 (65.7%) F:1821 (34.3%) NR:1 (0.0%) | M:5 (19.2%) F:21 (80.8%) | M:17 (23.0%) F:57 (77.0%) | M:22 (22.0%) F:78 (78.0%) | <0.001 |
| | F:1674 (33.6%) | | | | | | |
| | NR:1 (0.0%) | | | | | | |
| Mean age (SD) | 52.7 (11.8) | 49.4 (13.3) | 52.7 (11.8) | 52.7 (5.1) | 51.9 (8.7) | 52.1 (7.9) | 0.666 |
| Mean UKELD (SD) | 55.1 (5.6) | 51.0 (5.8) | 55.1 (5.6) | 49.2 (3.5) | 48.0 (3.1) | 48.4 (3.3) | <0.001 |
| Mean MELD (SD) | 17.0 (6.9) | 14.1 (6.6) | 17.0 (6.9) | 17.8 (5.9) | 15.3 (6.8) | 16.0 (6.7) | <0.001 |
| Mean Cr umol/l (SD) | 87.7 (52.5) | 101.8 (109.0) | 89.1 (59.0) | 264.0 (184.7) | 217.5 (187.4) | 316.3 (885.6) | <0.0001 |

PLD–Polycystic liver disease, VS–Variant syndrome pathway, CLD–chronic liver disease pathway, M–Male, F–Female, NR–Not Reported, SD–standard deviation

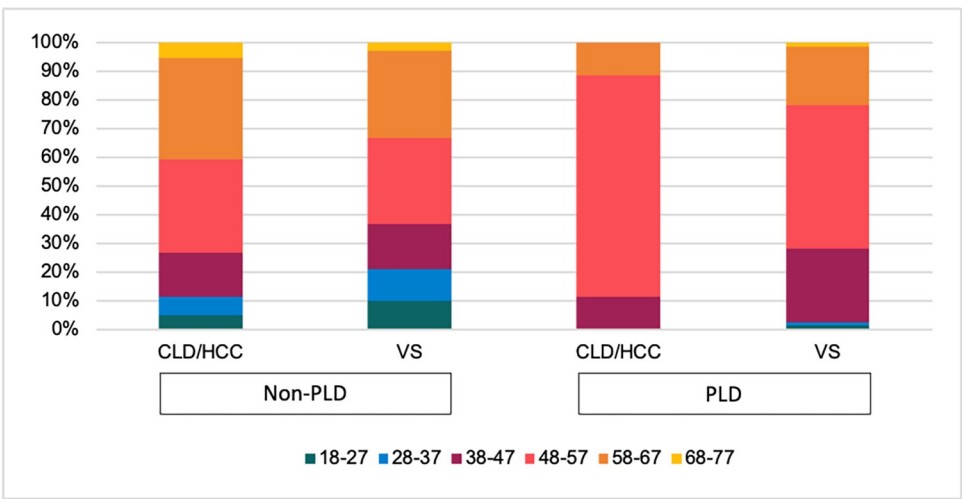

**Fig 2. Liver graft recipient age distribution for indication and transplant list type.**

non-PLD CLD/HCC, PLD CLD/HCC and other variant syndrome conditions (Fig 3). Overall transplant recipients with PLD indications had a mean waiting time for transplantation from listing of 508 days compared to 154 days for non-PLD indications.

## Patient and allograft survival

Overall, there was a low intraoperative mortality rate among liver allograft recipients with a total of 22 episodes over the study period (S2 Table). Patients with PLD undergoing

### Waiting times by liver transplant indication and list type

| List type | Non-PLD | | PLD | |
|---|---|---|---|---|
| | CLD/HCC | VS | CLD/HCC | VS |
| Mean (SD) days | 146.2 (188.7) | 276.1 (276.1) | 413.7 (499.7) | 541.5 (493.8) |
| Combined | 153.7 (197.2) | | 507.6 (496.0) | |

**Fig 3. Box and whisker plot of waiting time by elective liver transplant indication.**

**Table 2. Patient survival outcomes by transplantation indication and list type.**

| | | Patient survival | | |
|---|---|---|---|---|
| | | **30 days** | **1 year** | **5 years** |
| **Non-PLD** | **Total non-PLD (both)** | **4826/4943 (97.6%)** | **4544/4875 (93.2%)** | **2546/3329 (76.5%)** |
| | Liver only | 4784/4900 (97.6%) | 4504/4832 (93.2%) | 2527/3304 (76.5%) |
| | Liver-kidney | 42/43 (97.7%) | 40/43 (93.0%) | 19/25 (76.0%) |
| | **CLD/HCC (both)** | **4528/4637 (97.6%)** | **4272/4583 (93.2%)** | **2399/3134 (76.5%)** |
| | Liver only | 4497/4605 (97.7%) | 4241/4551 (93.2%) | 2384/3116 (76.5%) |
| | Liver-kidney | 31/32 (96.9%) | 31/32 (96.9%) | 15/18 (83.3%) |
| | **VS (both)** | **298/306 (97.4%)** | **272/292 (93.2%)** | **147/195 (75.4%)** |
| | Liver only | 287/295 (97.3%) | 263/281 (93.6%) | 143/188 (76.1%) |
| | Liver-kidney | 11/11 (100.0%) | 9/11 (81.8%) | 4/7 (57.1%) |
| **PLD** | **Total PLD (both)** | **94/100 (94.0%)** | **92/100 (92.0%)** | **52/64 (81.3%)** |
| | Liver only | 54/60 (90.0%) | 52/60 (86.7%) | 32/41 (78.0%) |
| | Liver-kidney | 39/39 (100.0%) | 39/39 (100.0%) | 20/23 (87.0%) |
| | **CLD/HCC (both)** | **25/26 (96.2%)** | **25/26 (96.2%)** | **13/14 (92.9%)** |
| | Liver only | 13/14 (92.9%) | 13/14 (92.9%) | 6/7 (85.7%) |
| | Liver-kidney | 11/11 (100.0%) | 11/11 (100.0%) | 7/7 (100.0%) |
| | **VS (both)** | **69/74 (93.2%)** | **67/74 (90.5%)** | **39/50 (78.0%)** |
| | Liver only | 41/46 (89.1%) | 39/46 (84.8%) | 26/34 (76.5%) |
| | Liver-kidney | 28/28 (100.0%) | 28/28 (100.0%) | 13/16 (81.3%) |

transplantation had comparable patient survival rates at 30 days and 1 year but higher survival rates at 5 years compared to those transplanted for non-PLD indications: 81.3% versus 76.5% (Tables 2 & 3, Fig 4). Comparable allograft survival rates were demonstrated at 30 days and 1 year but patients transplanted for PLD indications had higher 5-year allograft survival rates: 82.5% versus 77.3% (Tables 4 & 5, Fig 5).

Patients with PLD transplanted from the CLD/HCC list, compared to their counterparts transplanted from the VS list, had a higher patient survival rate at 30 days with 96.2% (95% CI 0.89–1.04) v 93.2% (95% CI 0.88–0.99); 1 year with 96.2% (95% CI 0.89–1.04) v 90.5% (95% CI 0.84–0.97); and 5 years with 92.9% (95% CI 0.89–1.04) vs 78.0% (95% CI0.75–0.93). Similar findings were demonstrated for allograft survival.

There were 860 reported episodes of liver allograft failure, excluding recipient death as the cause of failure (Table 6). Recurrent disease and vascular complications were the most commonly reported single cause of allograft failure in people with non-PLD indications for liver transplantation at 13.9% and 15.9% respectively. Acute vascular occlusion was the most commonly reported single cause of allograft failure in people with PLD indications for liver transplantation at 27.3% (3 patients).

**Table 3. Patient survival probability by transplantation indication and list type.**

| | | Patient survival probability (95% CI) | | |
|---|---|---|---|---|
| | | **30 days** | **1 year** | **5 years** |
| **Non-PLD** | **CLD/HCC** | 0.976 (0.97–0.98) | 0.933 (0.93–0.94) | 0.828 (0.82–0.84) |
| | **VS** | 0.974 (0.96–0.99) | 0.934 (0.91–0.96) | 0.818 (0.77–0.87) |
| **PLD** | **CLD/HCC** | 0.962 (0.89–1.04) | 0.962 (0.89–1.04) | 0.962 (0.89–1.04) |
| | **VS** | 0.932 (0.88–0.99) | 0.905 (0.84–0.97) | 0.838 (0.75–0.93) |
| Log rank (Mantel-Cox): Chi-square 4.385, df 3, P = 0.223 | | | | |

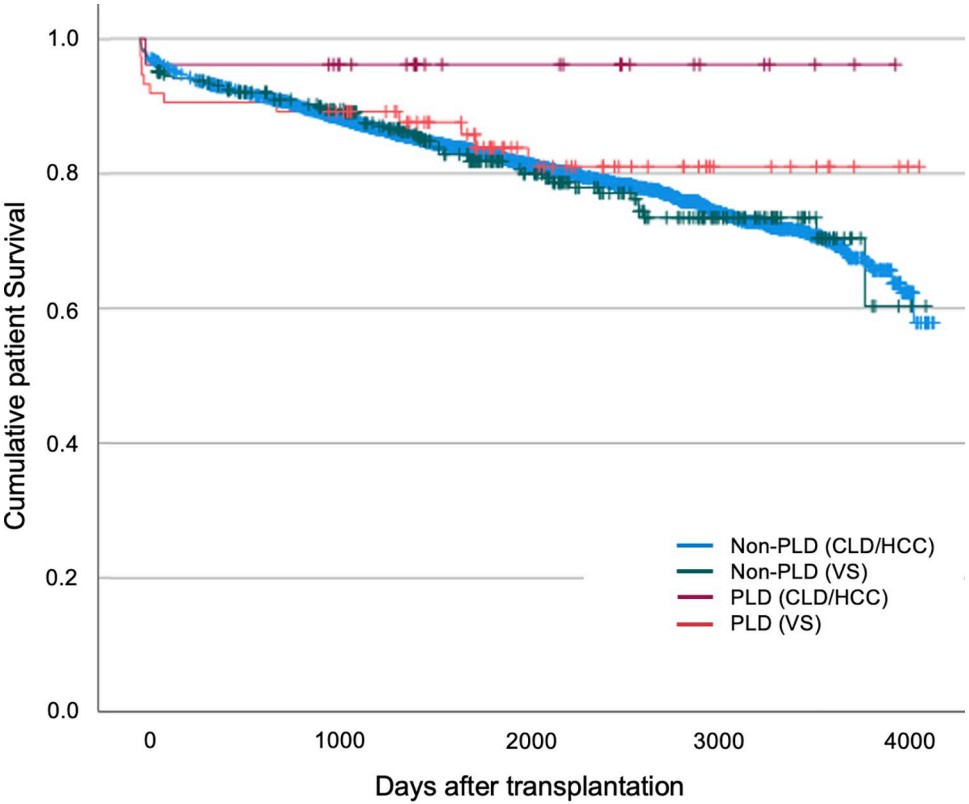

**Fig 4. Kaplan Meier survival curve for patient survival following elective transplantation.**

**Table 4. Allograft survival outcomes by transplantation indication and list type.**

| | | Allograft survival | | |
| --- | --- | --- | --- | --- |
| | | 30 days | 1 year | 5 years |
| Non-PLD | Total non-PLD (both) | 5003/5249 (95.3%) | 4630/5079 (91.2%) | 2521/3261 (77.3%) |
| | Liver only | 4956/5201 (95.3%) | 4586/5033 (91.1%) | 2498/3234 (77.2%) |
| | Liver-kidney | 47/48 (97.9%) | 44/46 (95.7%) | 22/26 (84.6%) |
| | CLD/HCC (both) | 4698/4931 (95.3%) | 4355/4781 (91.1%) | 2381/3080 (77.3%) |
| | Liver only | 4662/4894 (95.3%) | 4319/4744 (91.0%) | 2361/3057 (77.2%) |
| | Liver-kidney | 36/37 (97.3%) | 36/37 (97.3%) | 19/22 (86.4%) |
| | VS (both) | 305/318 (95.9%) | 275/298 (92.3%) | 140/181 (77.3%) |
| | Liver only | 294/307 (95.8%) | 267/289 (92.4%) | 137/177 (77.4%) |
| | Liver-kidney | 11/11 (100.0%) | 8/9 (88.9%) | 3/4 (75.0%) |
| PLD | Total PLD (both) | 93/99 (93.9%) | 91/99 (91.9%) | 52/63 (82.5%) |
| | Liver only | 53/59 (89.8%) | 51/59 (86.4%) | 32/41 (78.0%) |
| | Liver-kidney | 39/39 (100.0%) | 29/29 (100.0%) | 20/22 (90.9%) |
| | CLD/HCC (both) | 25/26 (96.2%) | 25/26 (96.2%) | 13/14 (92.9%) |
| | Liver only | 13/14 (92.9%) | 13/14 (92.9%) | 6/7 (85.7%) |
| | Liver-kidney | 11/11 (100.0%) | 11/11 (100.0%) | 7/7 (100.0%) |
| | VS (both) | 68/73 (93.2%) | 66/73 (90.4%) | 39/49 (79.6%) |
| | Liver only | 40/45 (88.9%) | 38/45 (84.4%) | 26/34 (76.5%) |
| | Liver-kidney | 28/28 (100.0%) | 28/28 (100.0%) | 13/15 (86.7%) |

**Table 5. Allograft survival probability by transplantation indication and list type.**

| | | Allograft survival probability (95% CI) | | |
|---|---|---|---|---|
| | | **30 days** | **1 year** | **5 years** |
| **Non-PLD** | **CLD/HCC** | 0.953 (0.89–1.01) | 0.913 (0.83–0.99) | 0.847 (0.75–0.95) |
| | **VS** | 0.959 (0.93–0.98) | 0.927 (0.90–0.96) | 0.858 (0.82–0.90) |
| **PLD** | **CLD/HCC** | 0.962 (0.89–1.04) | 0.962 (0.89–1.04) | 0.962 (0.89–1.04) |
| | **VS** | 0.932 (0.88–0.99) | 0.905 (0.84–0.97) | 0.857 (0.77–0.94) |
| | | Log rank (Mantel-Cox): Chi-square 3.866, df 3, P = 0.276 | | |

## Sub-group analysis of liver alone vs liver-kidney transplantation

A higher proportion of patients awaiting liver transplantation for PLD received a combined liver-kidney transplant compared to patients transplanted for other indications (Table 1).

Patients with PLD who were transplanted for VS indications had the longest waiting time for both liver allografts alone and combined liver-kidney transplantation (Fig 6).

In this study period, no patients receiving combined liver-kidney allografts were reported to have died intraoperatively (S2 Table). Survival rates for patients receiving liver only allografts and combined liver-kidney allografts are presented in Tables 2 and 4.

## Variation between transplant centres

There was an unexpected but statistically significant variation in the proportion of liver transplantations performed for PLD indications between the 6 liver transplant centres in the United

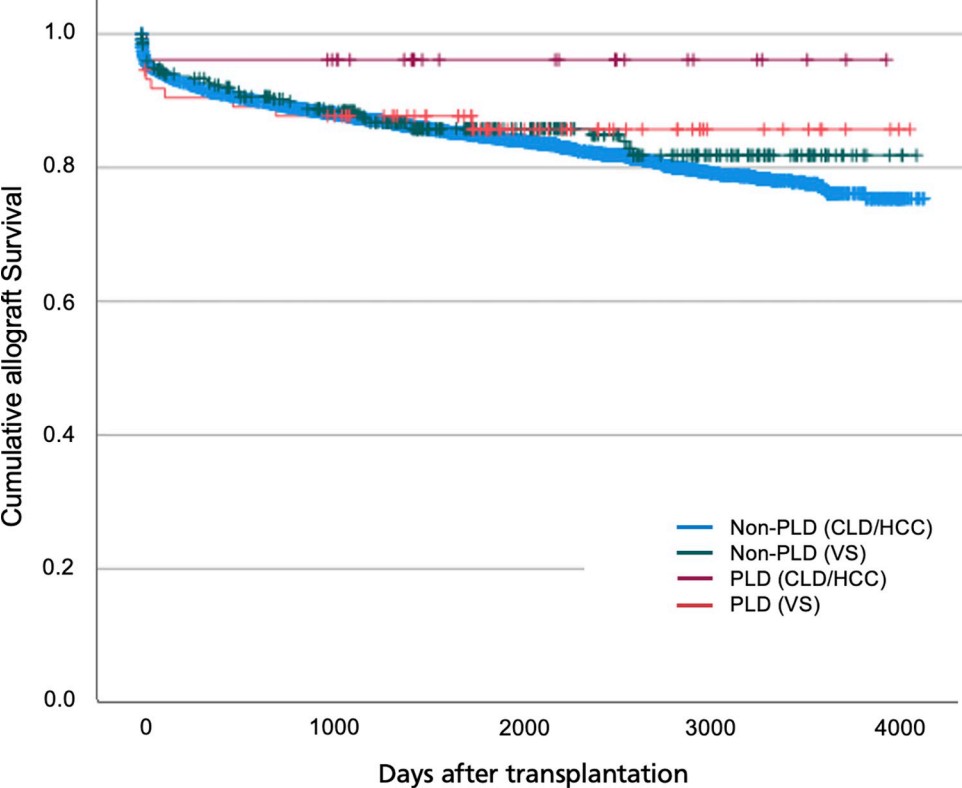

**Fig 5. Kaplan Meier survival curve for allograft survival following elective transplantation.**

**Table 6. Summary of allograft failure by PLD diagnosis and transplantation pathway.**

| | Non-PLD | | | PLD | | | |
|---|---|---|---|---|---|---|---|
| List type | CLD/HCC | VS | Total | CLD/HCC | VS | Total | Overall |
| Graft still functioning | 3803 (76.2%) | 247 (76.2%) | 4050 (76.2%) | 25 (96.2%) | 60 (81.1%) | 85 (85.0%) | 4135 (76.4%) |
| Recipient death | 434 (8.7%) | 34 (10.5%) | 468 (8.8%) | 0 (0.0%) | 4 (5.4%) | 4 (4.0%) | 472 (8.7%) |
| Primary non-functioning | 80 (10.0%) | 3 (0.9%) | 83 (1.6%) | 0 (0.0%) | 2 (2.7%) | 2 (2.0%) | 85 (1.6%) |
| Recurrent disease | 113 (2.3%) | 5 (1.5%) | 118 (2.2%) | 0 (0.0%) | 0 (0.0%) | 0 (0.0%) | 118 (2.2%) |
| All rejection | 76 (1.5%) | 4 (1.2%) | 80 (1.5%) | 0 (0.0%) | 1 (1.4%) | 1 (1.0%) | 81 (1.5%) |
| Acute Rejection | 16 (0.3%) | 0 (0.0%) | 16 (0.3%) | 0 (0.0%) | 0 (0.0%) | 0 (0.0%) | 16 (0.3%) |
| Chronic Rejection | 54 (1.1%) | 4 (1.2%) | 58 (1.1%) | 0 (0.0%) | 0 (0.0%) | 0 (0.0%) | 58 (1.1%) |
| Ductopenic rejection | 6 (0.1%) | 0 (0.0%) | 6 (0.1%) | 0 (0.0%) | 1 (1.4%) | 1 (1.0%) | 7 (0.1%) |
| All vascular | 123 (2.5%) | 12 (3.7%) | 135 (2.5%) | 1 (3.8%) | 2 (2.7%) | 3 (3.0%) | 138 (2.5%) |
| Acute vascular occlusion | 78 (1.6%) | 9 (2.8%) | 87 (1.6%) | 1 (3.8%) | 2 (2.7%) | 3 (3.0%) | 90 (1.7%) |
| Vascular occlusion | 34 (0.7%) | 2 (0.6%) | 36 (0.7%) | 0 (0.0%) | 0 (0.0%) | 0 (0.0%) | 36 (0.7%) |
| Non-thrombotic infarction | 11 (0.2%) | 1 (0.3%) | 12 (0.2%) | 0 (0.0%) | 0 (0.0%) | 0 (0.0%) | 12 (0.2%) |
| Biliary complications | 100 (2.0%) | 5 (1.5%) | 105 (2.0%) | 0 (0.0%) | 0 (0.0%) | 0 (0.0%) | 105 (1.9%) |
| Other | 167 (3.3%) | 8 (2.5%) | 175 (3.3%) | 0 (0.0%) | 3 (4.1%) | 3 (3.0%) | 178 (3.3%) |
| Unknown | 92 (1.8%) | 6 (1.9%) | 98 (1.8%) | 0 (0.0%) | 2 (2.7%) | 2 (2.0%) | 100 (1.8%) |
| **Total** | **4988** | **324** | **5312** | **26** | **74** | **100** | **5412** |

Kingdom (UK) which ranged from 0.3% to 2.7% of all liver transplants performed at each centre (P<0.0001), Table 7. Similarly, there was a statistically significant variation in the proportion of liver transplants being performed from the VS list ranging from 0.3% to 10.1% of the liver transplant performed at that centre (P<0.0001), Table 8. Conversely, there was less variation in the proportion of liver-kidney transplants performed at different centres, ranging from 0.9% to 2.6% (P = 0.1), Table 7.

## Discussion

Patients with PLD wait 3 times longer for liver transplantation than those with non-PLD CLD/HCC and other variant syndromes. This is independent of whether they are listed for CLD/HCC or VS indications. During the time period studied, only 1.5% of transplantations were for combined liver-kidney allografts. Longer waiting times can lead to advanced age,

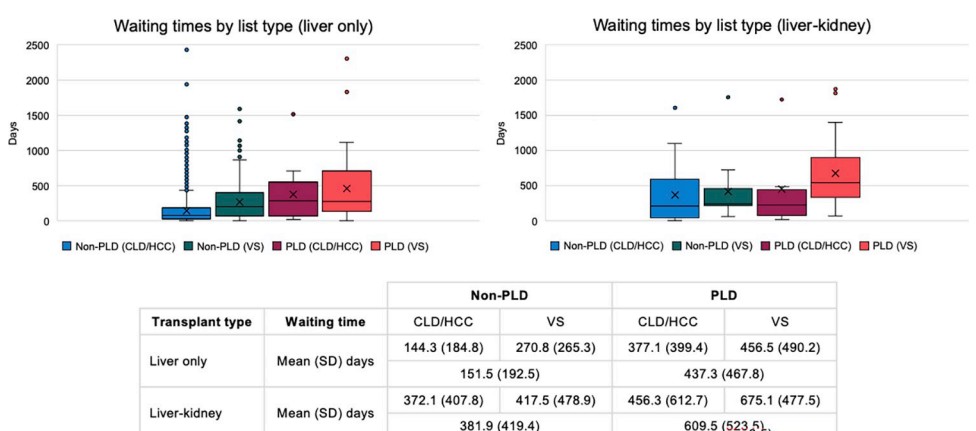

| | | Non-PLD | | PLD | |
|---|---|---|---|---|---|
| Transplant type | Waiting time | CLD/HCC | VS | CLD/HCC | VS |
| Liver only | Mean (SD) days | 144.3 (184.8) | 270.8 (265.3) | 377.1 (399.4) | 456.5 (490.2) |
| | | 151.5 (192.5) | | 437.3 (467.8) | |
| Liver-kidney | Mean (SD) days | 372.1 (407.8) | 417.5 (478.9) | 456.3 (612.7) | 675.1 (477.5) |
| | | 381.9 (419.4) | | 609.5 (523.5) | |

**Fig 6. Waiting time of elective liver and liver-kidney transplant allografts by indication and list type.**

**Table 7. Elective liver transplantation (PLD vs Non-PLD) by centre.**

| Centre | Allograft type | Non-PLD | | | PLD | | | |
|---|---|---|---|---|---|---|---|---|
| | | CLD/HCC | VS | Total | CLD/HCC | VS | Total | Overall |
| A | **Both** | **334 (99.1%)** | **0 (0.0%)** | **334 (99.1%)** | **2 (0.6%)** | **1 (0.3%)** | **3 (0.9%)** | **337 (100.0%)** |
| | Liver only | 331 (98.2%) | 0 (0.0%) | 331 (98.2%) | 0 (0.0%) | 0 (0.0%) | 0 (0.0%) | 331 (98.2%) |
| | Liver-kidney | 3 (0.9%) | 0 (0.0%) | 3 (0.9%) | 2 (0.6%) | 1 (0.3%) | 3 (0.9%) | 6 (1.8%) |
| B | **Both** | **880 (89.4%)** | **87 (8.8%)** | **967 (98.3%)** | **5 (0.5%)** | **12 (1.2%)** | **17 (1.7%)** | **984 (100.0%)** |
| | Liver only | 866 (88.0%) | 84 (8.5%) | 950 (96.5%) | 2 (0.2%) | 6 (0.6%) | 8 (0.8%) | 958 (97.4%) |
| | Liver-kidney | 14 (1.4%) | 3 (0.3%) | 17 (1.7%) | 3 (0.3%) | 6 (0.6%) | 9 (0.9%) | 26 (2.6%) |
| C | **Both** | **824 (95.0%)** | **40 (4.6%)** | **864 (99.7%)** | **0 (0.0%)** | **3 (0.3%)** | **3 (0.3%)** | **867 (100.0%)** |
| | Liver only | 820 (94.6%) | 37 (4.3%) | 857 (98.8%) | 0 (0.0%) | 2 (0.2%) | 2 (0.2%) | 859 (99.1%) |
| | Liver-kidney | 4 (0.5%) | 3 (0.3%) | 7 (0.8%) | 0 (0.0%) | 1 (0.1%) | 1 (0.1%) | 8 (0.9%) |
| D | **Both** | **770 (94.2%)** | **33 (4.0%)** | **803 (98.3%)** | **5 (0.6%)** | **9 (1.1%)** | **14 (1.7%)** | **817 (100.0%)** |
| | Liver only | 767 (93.9%) | 33 (4.0%) | 800 (97.9%) | 3 (0.4%) | 3 (0.4%) | 6 (0.7%) | 806 (98.7%) |
| | Liver-kidney | 3 (0.4%) | 0 (0.0%) | 3 (0.4%) | 2 (0.2%) | 6 (0.7%) | 8 (1.0%) | 11 (1.3%) |
| E | **Both** | **1438 (90.1%)** | **117 (7.3%)** | **1555 (97.4%)** | **7 (0.4%)** | **34 (2.1%)** | **41 (2.6%)** | **1596 (100.0%)** |
| | Liver only | 1426 (89.3%) | 114 (7.1%) | 1540 (96.5%) | 5 (0.3%) | 25 (1.6%) | 30 (1.9%) | 1570 (98.4%) |
| | Liver-kidney | 12 (0.8%) | 3 (0.2%) | 15 (0.9%) | 2 (0.1%) | 9 (0.6%) | 11 (0.7%) | 26 (1.6%) |
| F | **Both** | **742 (91.5%)** | **47 (5.8%)** | **789 (97.3%)** | **7 (0.9%)** | **15 (1.8%)** | **22 (2.7%)** | **811 (100.0%)** |
| | Liver only | 738 (91.0%) | 45 (5.5%) | 783 (96.5%) | 4 (0.5%) | 10 (1.2%) | 14 (1.7%) | 797 (98.3%) |
| | Liver-kidney | 4 (0.5%) | 2 (0.2%) | 6 (0.7%) | 3 (0.4%) | 5 (0.6%) | 8 (1.0%) | 14 (1.7%) |
| Total | **Both** | **4988 (92.2%)** | **324 (6.0%)** | **5312 (98.2%)** | **26 (0.5%)** | **74 (1.4%)** | **100 (1.8%)** | **5412 (100.0%)** |
| | Liver only | 4948 (91.4%) | 313 (5.8%) | 5261 (97.2%) | 14 (0.3%) | 46 (0.8%) | 60 (1.1%) | 5321 (98.3%) |
| | Liver-kidney | 40 (0.7%) | 11 (0.2%) | 51 (0.9%) | 12 (0.2%) | 28 (0.5%) | 40 (0.7%) | 91 (1.7%) |

malnutrition and increased frailty which could lead to patients dying on the waiting list, becoming unsuitable for transplantation and experiencing a worse quality of life [25, 26]. Our study demonstrates that there are comparable survival rates, patient and allograft, for patients with PLD and non-PLD indications for transplantation at 30 days and 1 year. However, transplant recipients with PLD have higher 5-year survival rates.

Our study also demonstrates a difference between patients with PLD transplanted from the two different list types, CLD/HCC and VS. PLD patients who received a liver allograft from the CLD/HCC list had greater patient, allograft and transplant survival rates at 30 days, 1 year and 5 years. One potential explanation could be the shorter waiting time for the patients transplanted from the CLD/HCC list.

The current ranking of patients on the waiting list for variant syndrome indications is based solely on time on the waiting list. This does not take into consideration the impact on functional life, intractable pain and severely diminished quality of life [27]. Lang et al reported symptomatic relief in all patients following transplantation [28]. Furthermore, Kirchner et al

**Table 8. Elective liver transplantation (CLD/HCC vs VS) by centre.**

| List type | Centre | | | | | | |
|---|---|---|---|---|---|---|---|
| | A | B | C | D | E | F | Total |
| CLD/HCC | 336 (99.7%) | 885 (89.9%) | 824 (95.0%) | 775 (94.9%) | 1445 (90.5%) | 749 (92.4%) | 5014 (92.6%) |
| VS | 1 (0.3%) | 99 (10.1%) | 43 (5.0%) | 42 (5.1%) | 151 (9.5%) | 62 (7.6%) | 398 (7.4%) |
| Total | 337 | 984 | 867 | 817 | 1596 | 811 | 5412 |
| % of UK transplants (2010–2017) | 6.2% | 18.2% | 16.0% | 15.1% | 29.5% | 15.0% | 100.0% |

demonstrated an improved quality of life following transplantation with 91% of their patients reporting feeling "much better" or "better" based on the Short Form Health Survey, SF-36 [9]. Several authors have recommended that patients should not wait until the onset of end-stage complications of PLD before being offered the option of transplantation [22, 28, 29]. According to current indications, patients in the UK with PLD are waiting until their symptoms are "intractable".

Our results are specific to the UK population of people with PLD receiving a liver transplant within the specific time period. Although organ allocation policies vary between different countries limiting generalisability, this study adds to the existing literature from other countries demonstrating similar outcomes. An increased risk of perioperative mortality reported in the literature is suspected to be due to the increased operative complexity in patients with PLD. This is supported by our dataset where 1% of PLD recipients died intraoperatively compared to 0.4% of non-PLD recipients. This could be made worse by delaying transplantation due to the continued increase in cyst and liver size [26, 30]. Nonetheless, our study demonstrates that patients with PLD have a better patient and liver allograft survival rate following transplantation independent of whether transplanted for CLD/HCC or VS indications. Other single centre studies have reported similar findings (86–96% survival) [31, 32].

The 860 reported episodes of allograft failure represent 15.9% of transplantation episodes within the time period in our study. This is consistent with worldwide reported liver allograft failure rates of 5–22% [33]. Recurrence of primary disease was the most common cause of liver allograft failure amongst patients with non-PLD indications for liver transplantation. As would be expected, there was no evidence of recurrent disease amongst recipients with PLD given that liver transplantation is curative for their genetic condition. Vascular complications were the most common cause of liver allograft failure in recipients with both non-PLD and PLD indications. This is in line with other studies which report vascular complications as a frequent indication for re-transplantation [33–35].

An unexpected finding from our study was the significant variation in the rate of liver transplantation for PLD patients between transplant centres in the UK. This may be explained by the significant variation between centres in the proportion of transplanted allografts from the variant syndrome list. Another reason could be the interpretation of the criteria in the current liver allocation policy used to inform listing practices. The term "intractable" is used to quantify the severity of symptoms that would warrant referral for liver transplantation which is clearly subjective. The criteria also includes "symptom(s) due to (the) mass of liver or pain unresponsive to cystectomy" although cystectomy is not widely implemented in UK clinical practice [36]. Clarifying the precise indications is likely to promote more equitable access to liver transplantation for PLD patients across the UK.

## Strengths and limitations

We report transplantation rates prior to the updated listing policy in 2018 for people with and without PLD in UK-based transplant centres. One weakness of this study is that we are unable to comment on the characteristics of patients not listed for transplantation or those on the waiting list. Although there was overall a high level of complete data return, there was incomplete reporting of intraoperative death (87.9% complete), patient death (90.7% complete) and graft failure (99.6% complete).

## Recommendations for clinical practice and future research

We recommend further research to assess the impact of the change from regional to national liver allograft allocation since 2018. These findings could further refine the current national

allocation policy, especially for patients with a variant syndrome indication. Patients on the waiting list should be studied longitudinally after listing and outcome data obtained on those assessed for listing but who are then not listed. Finally, the variation in transplantation rates for patients with PLD between different UK centres should be analysed.

## Conclusions

Patients with PLD have a longer waiting time to transplantation compared to those with other indications for liver transplantation but better long-term patient and liver allograft survival rates. This is independent of the type of list at registration, chronic liver disease/hepatocellular carcinoma or variant syndrome list. There is significant variation in the numbers of patients with PLD undergoing liver transplantation at different UK-based transplant centres.

## Supporting information

**S1 Checklist. STrengthening the Reporting of OBservational studies in Epidemiology (STROBE) checklist.**
(DOCX)

**S1 Table. Patient ethnicity by PLD diagnosis and transplantation pathway.**
(PNG)

**S2 Table. Intraoperative death.**
(PNG)

## Acknowledgments

National Institute for Health and Care Research, NHS Blood and Transplantation, statistics services and the transplant patients who consented to data collection.

## Author Contributions

**Conceptualization:** Matt Gittus, Joanna Moore, Albert C. M. Ong.

**Data curation:** Matt Gittus.

**Formal analysis:** Matt Gittus.

**Methodology:** Matt Gittus.

**Project administration:** Matt Gittus.

**Supervision:** Joanna Moore, Albert C. M. Ong.

**Writing – original draft:** Matt Gittus.

**Writing – review & editing:** Matt Gittus, Joanna Moore, Albert C. M. Ong.

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
