## [Decision Letter · Decision Letter 0]

26 Jul 2023

PONE-D-23-17780Liver transplant recipients with polycystic liver disease have longer waiting times but better long-term clinical outcomes than those with liver disease due to other causes: A retrospective cross-sectional studyPLOS ONE

Dear Dr. Gittus,

Thank you for submitting your manuscript to PLOS ONE. After careful consideration, we feel that it has merit but does not fully meet PLOS ONE’s publication criteria as it currently stands. Therefore, we invite you to submit a revised version of the manuscript that addresses the points raised during the review process.

We look forward to receiving your revised manuscript.

Kind regards,

Stanislaw Stepkowski

Academic Editor

PLOS ONE

Journal Requirements:

3. Please ensure that you refer to Figure 6 in your text as, if accepted, production will need this reference to link the reader to the figure.

Additional Editor Comments:

PONE-D-23-17780

The Editor reviewed the manuscript, and agreed with the Reviewer # 1. However, additional consideration need to be addressed in the manuscript.

The Abstract needs to be enriched with the actual specific data. The Results section in the Abstract is descriptive. Please provide specific numbers for the waiting time and for the survival. It is also confusing in precision of describing patients who underwent a combined liver-kidney transplantation. Out of 100 recipients with PLD how many had liver and how many had liver-kidney transplantation? Are there their waiting times different? What about their graft and patient survival times at 30 days, one and five years? The last section should be entitled conclusions.

The authors need to explain better the reasons for the graft failure in PLD group vs. non-PLD group as this point is very interesting. The possible difference between two groups may be related to the recurrence of the disease in non-PLD group, but this needs to be better documented.

Please address the question about the mortality on the waiting list for PLD group as this is especially interesting with an extended waiting time for the transplant. As suggested by the Reviewer # 1, please present more extensively about the waiting time mortality in PLD and non-PLD groups: additional table may be helpful.

Please address the comments by the Reviewer # 1:

Lines 211-213: Discussed overall allograft failure in the data. The graph shows that allograft failure is much less in PCLD and that there is no recurrent disease in these patients, which is the leading reason for allograft failure in non-PCLD transplants. I recommend highlighting this point in your text.

Lines 250-253: It would be interesting to see data on wait list mortality for these patients, especially in light of PCLD-VS patients having the longest wait time. If you have that data, I would recommend presenting it in this paper.

They report several interesting findings. They find that patients transplanted for PCLD have a higher 5 year survival but a longer wait list time. They found that PCLD patients listed for variant syndrome had the longest waiting time- longer than other variant syndrome patients and longer than PCLD listed with HCC or CLD. They also find that there is a significant difference between transplant centers, suggesting that the VS criteria in particular can lead to variations in interpretation.

This study provides insight into the transplant outcomes for patients with PCLD.

All changes must be explained and marked in the revised vesrion.

Reviewers' comments:

Reviewer's Responses to Questions

**Comments to the Author**

1. Is the manuscript technically sound, and do the data support the conclusions?

Reviewer #1: Yes

2. Has the statistical analysis been performed appropriately and rigorously? 

Reviewer #1: Yes

3. Have the authors made all data underlying the findings in their manuscript fully available?

Reviewer #1: Yes

4. Is the manuscript presented in an intelligible fashion and written in standard English?

Reviewer #1: Yes

5. Review Comments to the Author

Reviewer #1: This paper reviews the transplant data for patients listed for transplant with PCLD, both with liver dysfunction and those listed for variant syndrome (intractable symptoms from mass effect or portal hypertension). The authors analyzed the transplant outcomes of patients with PCLD compared to those transplanted for other indications.

They report several interesting findings. They find that patients transplanted for PCLD have a higher 5 year survival but a longer wait list time. They found that PCLD patients listed for variant syndrome had the longest waiting time- longer than other variant syndrome patients and longer than PCLD listed with HCC or CLD. They also find that there is a significant difference between transplant centers, suggesting that the VS criteria in particular can lead to variations in interpretation.

This study provides insight into the transplant outcomes for patients with PCLD.

Lines 211-213: Discussed overall allograft failure in the data. The graph shows that allograft failure is much less in PCLD and that there is no recurrent disease in these patients, which is the leading reason for allograft failure in non-PCLD transplants. I recommend highlighting this point in your text.

Lines 250-253: It would be interesting to see data on wait list mortality for these patients, especially in light of PCLD-VS patients having the longest wait time. If you have that data, I would recommend presenting it in this paper.

6. PLOS authors have the option to publish the peer review history of their article (what does this mean?). If published, this will include your full peer review and any attached files.

Reviewer #1: No

---

## [Author Response · Author response to Decision Letter 0]

11 Aug 2023

Journal Requirements:

Thank you for this recommendation and the link to your guideline on style requirements.

2. PLOS requires an ORCID iD for the corresponding author in Editorial Manager on papers submitted after December 6th, 2016. Please ensure that you have an ORCID iD and that it is validated in Editorial Manager. 

Thank you, the ORCID iD has been added to the submission on the Editorial Manager.

3. Please ensure that you refer to Figure 6 in your text as, if accepted, production will need this reference to link the reader to the figure.

Thank you for highlighting this error. It has been adjusted in the revised manuscript.

Additional Editor Comments:  

PONE-D-23-17780 The Editor reviewed the manuscript, and agreed with the Reviewer # 1. However, additional consideration need to be addressed in the manuscript.  The Abstract needs to be enriched with the actual specific data. The Results section in the Abstract is descriptive. Please provide specific numbers for the waiting time and for the survival. It is also confusing in precision of describing patients who underwent a combined liver-kidney transplantation. Out of 100 recipients with PLD how many had liver and how many had liver-kidney transplantation? Are there their waiting times different? What about their graft and patient survival times at 30 days, one and five years? The last section should be entitled conclusions.

Thank you for your recommendation. We have increased the amount of data included in the abstract in the results section. The data points suggested for inclusion have been specifically included in the revised abstract. As requested the last section has been entitled “conclusions”.

The authors need to explain better the reasons for the graft failure in PLD group vs. non-PLD group as this point is very interesting. The possible difference between two groups may be related to the recurrence of the disease in non-PLD group, but this needs to be better documented.  Please address the question about the mortality on the waiting list for PLD group as this is especially interesting with an extended waiting time for the transplant. As suggested by the Reviewer # 1, please present more extensively about the waiting time mortality in PLD and non-PLD groups: additional table may be helpful.

Thank you for this recommendation, it has been addressed through improving the data granularity in all tables included in the manuscript. As you suggested the main difference between the non-PLD and PLD groups is related to recurrence of disease in the non-PLD group. This is important as the PLD group cannot have recurrence due to the genetic basis for their condition. The updated section of the results section is located on lines 234-239 of the revised manuscript. Unfortunately, we do not currently have the data in regard to patients on the waiting list so cannot provide any analysis on mortality whilst on the waiting list. We are planning to request this data for a future study. 

Please address the comments by the Reviewer # 1: 

Lines 211-213: Discussed overall allograft failure in the data. The graph shows that allograft failure is much less in PCLD and that there is no recurrent disease in these patients, which is the leading reason for allograft failure in non-PCLD transplants. I recommend highlighting this point in your text. 

Thank you for your recommendation, we agree that this point needs to made more clearly in the manuscript. This has been highlighted on lines 333-342 which has been revised to include the comment that there is no recurrent disease in PLD patients due to the nature of their disease. 

Lines 250-253: It would be interesting to see data on wait list mortality for these patients, especially in light of PCLD-VS patients having the longest wait time. If you have that data, I would recommend presenting it in this paper.

Thank you, we agree that this would be a very interesting approach to highlight the potential harm from increased waiting times in the PLD group. We are planning to request this data for a future study.

They report several interesting findings. They find that patients transplanted for PCLD have a higher 5 year survival but a longer wait list time. They found that PCLD patients listed for variant syndrome had the longest waiting time- longer than other variant syndrome patients and longer than PCLD listed with HCC or CLD. They also find that there is a significant difference between transplant centers, suggesting that the VS criteria in particular can lead to variations in interpretation. This study provides insight into the transplant outcomes for patients with PCLD.  All changes must be explained and marked in the revised vesrion.   

Thank you for these comments. Based on your comments we have improved the data granularity for this section by revising table 5 and including table 6. This hopefully indicates the potential role that transplantation rates from the VS list can have on variation between centres. This has also been indicated in the discussion on lines 345-347.

---

## [Editor Report · Decision Letter 1]

7 Nov 2023

Liver transplant recipients with polycystic liver disease have longer waiting times but better long-term clinical outcomes than those with liver disease due to other causes: A retrospective cross-sectional study

PONE-D-23-17780R1

Dear Dr. Gittus,

We’re pleased to inform you that your manuscript has been judged scientifically suitable for publication and will be formally accepted for publication once it meets all outstanding technical requirements.

Kind regards,

Yuri Boteon, M.D., Ph.D.

Academic Editor

PLOS ONE

Additional Editor Comments (optional):

The authors have done a commendable job by conducting a comprehensive investigation involving a large number of patients over an extensive period. After carefully reviewing the revised manuscript along with the previous reviewer's and editor's feedback, I am pleased to report that the authors have addressed all the comments and improved the manuscript accordingly. Therefore, I am confident that the manuscript meets the publication criteria of Plos One, has significant value for publication, and can be accepted in its current form.
---

## [Editor Report · Acceptance letter]

13 Nov 2023

PONE-D-23-17780R1 

Liver transplant recipients with polycystic liver disease have longer waiting times but better long-term clinical outcomes than those with liver disease due to other causes: A retrospective cross-sectional study 

Dear Dr. Gittus:

I'm pleased to inform you that your manuscript has been deemed suitable for publication in PLOS ONE. Congratulations! Your manuscript is now with our production department. 

Kind regards, 

on behalf of

Prof. Yuri Longatto Boteon 

Academic Editor

PLOS ONE